# Evidence for persistence of the SHIV reservoir early after MHC haploidentical hematopoietic stem cell transplantation

Lucrezia Colonna[1,2], Christopher W. Peterson[3,4], John B. Schell [1,11], Judith M. Carlson[1], Victor Tkachev [1], Melanie Brown[1], Alison Yu[1], Sowmya Reddy[3], Willi M. Obenza[3], Veronica Nelson[3], Patricia S. Polacino[5], Heather Mack[5], Shiu-Lok Hu[5,6], Katie Zeleski[1], Michelle Hoffman[1], Joe Olvera[5], Scott N. Furlan[1,2], Hengqi Zheng [1,2], Agne Taraseviciute[1,2], Daniel J. Hunt[1], Kayla Betz[1], Jennifer F. Lane[5,7], Keith Vogel[5], Charlotte E. Hotchkiss[5], Cassie Moats[5], Audrey Baldessari[5], Robert D. Murnane[5], Christopher English[5], Cliff A. Astley[5], Solomon Wangari[5], Brian Agricola[5], Joel Ahrens[5], Naoto Iwayama[5], Andrew May[5], Laurence Stensland[8], Meei-Li W. Huang[8], Keith R. Jerome[8,9], Hans-Peter Kiem [3,4,10] & Leslie S. Kean[1,2,4,12]

Allogeneic transplantation (allo-HCT) has led to the cure of HIV in one individual, raising the question of whether transplantation can eradicate the HIV reservoir. To test this, we here present a model of allo-HCT in SHIV-infected, cART-suppressed nonhuman primates. We infect rhesus macaques with SHIV-1157ipd3N4, suppress them with cART, then transplant them using MHC-haploidentical allogeneic donors during continuous cART. Transplant results in ~100% myeloid donor chimerism, and up to 100% T-cell chimerism. Between 9 and 47 days post-transplant, terminal analysis shows that while cell-associated SHIV DNA levels are reduced in the blood and in lymphoid organs post-transplant, the SHIV reservoir persists in multiple organs, including the brain. Sorting of donor-vs.-recipient cells reveals that this reservoir resides in recipient cells. Moreover, tetramer analysis indicates a lack of virus-specific donor immunity post-transplant during continuous cART. These results suggest that early post-transplant, allo-HCT is insufficient for recipient reservoir eradication despite high-level donor chimerism and GVHD.

---

[1] Ben Towne Center for Childhood Cancer Research, Seattle Children's Research Institute, 1100 Olive Way, Suite 100, Seattle, WA 98101, USA. [2] Department of Pediatrics, University of Washington, Box 3563201959 NE Pacific Street, Seattle, WA 98195-6320, USA. [3] Stem Cell and Gene Therapy Program, Clinical Research Division, Fred Hutchinson Cancer Research Center, 1100 Fairview Ave N, Seattle 98109 WA, USA. [4] Department of Medicine, University of Washington, Box 3564201959 NE Pacific Street, Seattle, WA 98195-6420, USA. [5] Washington National Primate Research Center, 1705 NE Pacific StreetBox 357330Seattle, WA 98195, USA. [6] Department of Pharmaceutics, University of Washington, Box 3576101959 NE Pacific Street, Seattle, WA 98195-7610, USA. [7] Montana State University, 100 ARC Bldg. Tietz Hall, Montana State University, P.O. Box 173640Bozeman, MT 59717, USA. [8] Department of Laboratory Medicine, University of Washington, Box 3571101959 NE Pacific Street, Seattle, WA 98195-7110, USA. [9] Vaccine and Infectious Disease Division, Fred Hutchinson Cancer Research Center, 1100 Fairview Ave NMail Stop E5-110Seattle, WA 98109, USA. [10] Department of Pathology, University of Washington, Box 3574701959 NE Pacific Street, Seattle, WA 98195-7470, USA. [11]Present address: Vaccine and Gene Therapy Institute, Oregon Health & Science University, 505 NW 185th Ave, Beaverton OR 97006, USA. [12]Present address: Boston Children's Hospital and the Dana-Farber Cancer Institute, Karp Research Building, Room 08215, 1 Blackfan Circle, Boston 02115 MA, USA. These authors contributed equally: Lucrezia Colonna, Christopher W. Peterson, John B. Schell.  Correspondence and requests for materials should be addressed to L.S.K. (email: leslie.kean@childrens.harvard.edu)

Despite notable advances in the development of combination antiretroviral therapy (cART) for long-term suppression of HIV-1 viremia, a strategy capable of suppressing viral replication in the absence of cART remains elusive. As such, patients must adhere to a financially burdensome lifelong regimen of cART; withdrawal typically results in viral rebound 1–4 weeks post cART interruption[1,2]. Although multiple approaches are currently under investigation to induce cART-free virus remission, allogeneic stem cell transplantation (allo-HCT) with CCR5-null (CCR5Δ32) cells has led to the only documented cure to date, the Berlin patient[3,4]. Although allo-HCT is not practical in most HIV+ patients, it is feasible and necessary in those with associated hematological malignancies. Indeed, HIV+ patients are at increased risk for development of cancers, including Hodgkin and non-Hodgkin lymphomas[5], acute leukemias[6], myelodysplastic syndromes[7], as well as solid tumors of the lung, bladder, and gut[5,8].

Given that chemotherapy-refractory hematologic malignancies are the most common cause of cancer-related deaths in HIV+ patients[9], these patients are strong candidates for allo-HCT, which led to the striking results seen in the Berlin patient. However, the requirements of MHC matching combined with the rarity of CCR5Δ32 donors[10,11] make these donors difficult to find, and even when available, subsequent attempts to cure HIV infection in this population have been unsuccessful[12,13]. Henrich et al. described two HIV-1+ patients, known as the Boston Patients A and B, who developed Hodgkin lymphoma and myelodysplastic syndrome, respectively, and received allogeneic cell products from CCR5 wild type donors following a reduced-intensity pre-transplant conditioning regimen[14,15]. Both patients were maintained on continuous cART for 4.3 and 2.6 years, respectively, after transplant, during which time no viral DNA was detected in the patients' PBMC by sensitive qPCR assays[15]. However, after an analytic treatment interruption (ATI), plasma viremia rebounded in both patients, 12–32 weeks after cART was discontinued[15]. These results indicated that allogeneic HSCT without HIV-resistant stem cells reduced, but did not eradicate, the HIV reservoir in these two patients. These data raise the critical questions of which anatomic reservoir locations are resistant to allo-HCT (a question difficult, if not impossible, to address in clinical studies), whether the reservoir spreads from recipient to donor when transplant occurs in the presence of cART, and strongly suggests that viral resistance factors may be necessary to protect donor cells from becoming infected.

We have previously shown in nonhuman primate (NHP) modeling experiments that transplantation with unmodified autologous hematopoietic stem cells (HSCs) is insufficient to achieve cART-free virus remission[16,17]. We first used simian/human immunodeficiency virus carrying an HIV-1 reverse transcriptase (RT-SHIV) to infect rhesus macaques, followed by suppressive cART[16]. In this study, 2/3 animals rebounded in the peripheral blood following transplantation and withdrawal of cART. In the third animal, although viremia remained stably suppressed in peripheral blood at necropsy, tissue-associated viral DNA was later recovered. More recently, 100% of pigtail macaques infected with an HIV-enveloped SHIV, SHIV-1157ipd3N4 (SHIV-C) also rebounded following autologous transplantation[17]. Interestingly, we found that transplanted animals displayed a significant increase in plasma and tissue viral rebound relative to controls, suggesting that the non-specific impact of the myeloablative conditioning regimen on virus-specific immune cells may offset its benefit in killing virus-infected cells. These large animal studies are consistent with clinical data, which suggest that autologous transplantation with unmodified stem cells in suppressed HIV+ patients and continuing cART administration is safe, but is not curative[18–20].

In the setting of allo-HCT, significant ongoing efforts focus on harnessing a Graft vs. Leukemia effect (GVL) in patients with hematological malignancies[21]. Whether such an analogous mechanism might exist against latent virus in HIV+ patients (Graft vs. Viral Reservoir, or 'GVVR'), i.e. whether activated donor T cells might promote the clearance of host infected cells with increased efficiency compared to autologous T cells, remains unclear. Indeed, the Berlin patient developed graft vs. host disease (GVHD) post-transplant[22], a complication often linked to GVL, which may have contributed to the clearance of residual infected host cells.

To better understand the kinetics of viral reservoir size in suppressed HIV+ patients following allo-HCT, we have now performed the first allo-HCT experiments in SHIV-infected, cART-treated rhesus macaques. These animals were extensively monitored for SHIV-C RNA and DNA in peripheral and tissue reservoirs before and after transplant, and for the recipient vs. donor origin of the DNA reservoir at necropsy. Our goal was to create a comprehensive temporal and anatomical map of the SHIV reservoir, to better understand sites of viral persistence that may have contributed to rebound in the Boston patients, but may have been eradicated in the Berlin Patient. These studies are directly applicable to the growing population of HIV malignancy patients for whom allo-HCT is a first line treatment, and offer insights into mechanisms of reservoir control that could also be applied to otherwise healthy HIV+ patients. Our study indicates that allogeneic HCT in the absence of HIV resistance factors is not sufficient to eradicate the residual SHIV DNA reservoir in recipient cells early after transplant, despite high levels of donor chimerism, and GVHD.

## Results

**SHIV infection and reservoir analysis before and after cART.** Six rhesus macaques (RM) were infected IV with SHIV-C as previously described[23], then monitored without cART for six months to allow rigorous reservoir establishment and to characterize the viral set point in each animal. At 6 months post-infection, cART (tenofovir, emtricitabine, raltegravir) was initiated[23] and continued for the remainder of the experiment. Plasma viral loads (PVL) were monitored bi-weekly by quantitative real-time PCR[24]. In agreement with published results[25], all animals showed peak plasma viral loads within 2 weeks of infection, with an average of $1.46 \times 10^7 \pm 0.19 \times 10^7$ copies/mL, 10–11 days post infection. Viral loads subsequently declined, and reached a new set-point (with an average viral load measured at Day 140–141 post-infection of $4.9 \times 10^4 \pm 2.03 \times 10^4$ copies/mL; Fig. 1a). cART treatment was initiated between Day 184–189 post-infection, and resulted in a rapid decline in plasma viral RNA, to below the limit of detection (<30 copies/mL), ~4 weeks after treatment initiation. Consistent with previously published studies in both RM[26] and humans[27], and with the development of an anti-SHIV serologic response, all animals developed high titers of whole virus-reactive antibodies (Fig. 1b), with peak average titers in the range of $10^4$–$10^5$ detected between days 100–200 post-infection. As expected[23,28], antibody levels declined post cART treatment, with titers of $10^3$–$10^4$ after cART initiation (Fig. 1b). In addition to anti-SHIV antibodies, T cell counts as well as hemoglobin and platelet counts were monitored in infected/cART-treated animals (Fig. 1c–f). Virus-dependent decreases in peripheral blood CD4 + T cell counts were small but statistically significant at 6 months post-infection, prior to cART initiation ($544 \pm 129 \times 10^3/\mu L$ compared to $768 \pm 122 \times 10^3/\mu L$ pre-infection, $p = 0.034$), and normalized after cART treatment (Fig. 1c). As expected, CD8 + T cell counts were not significantly altered by SHIV-C infection or cART (Fig. 1d). Most

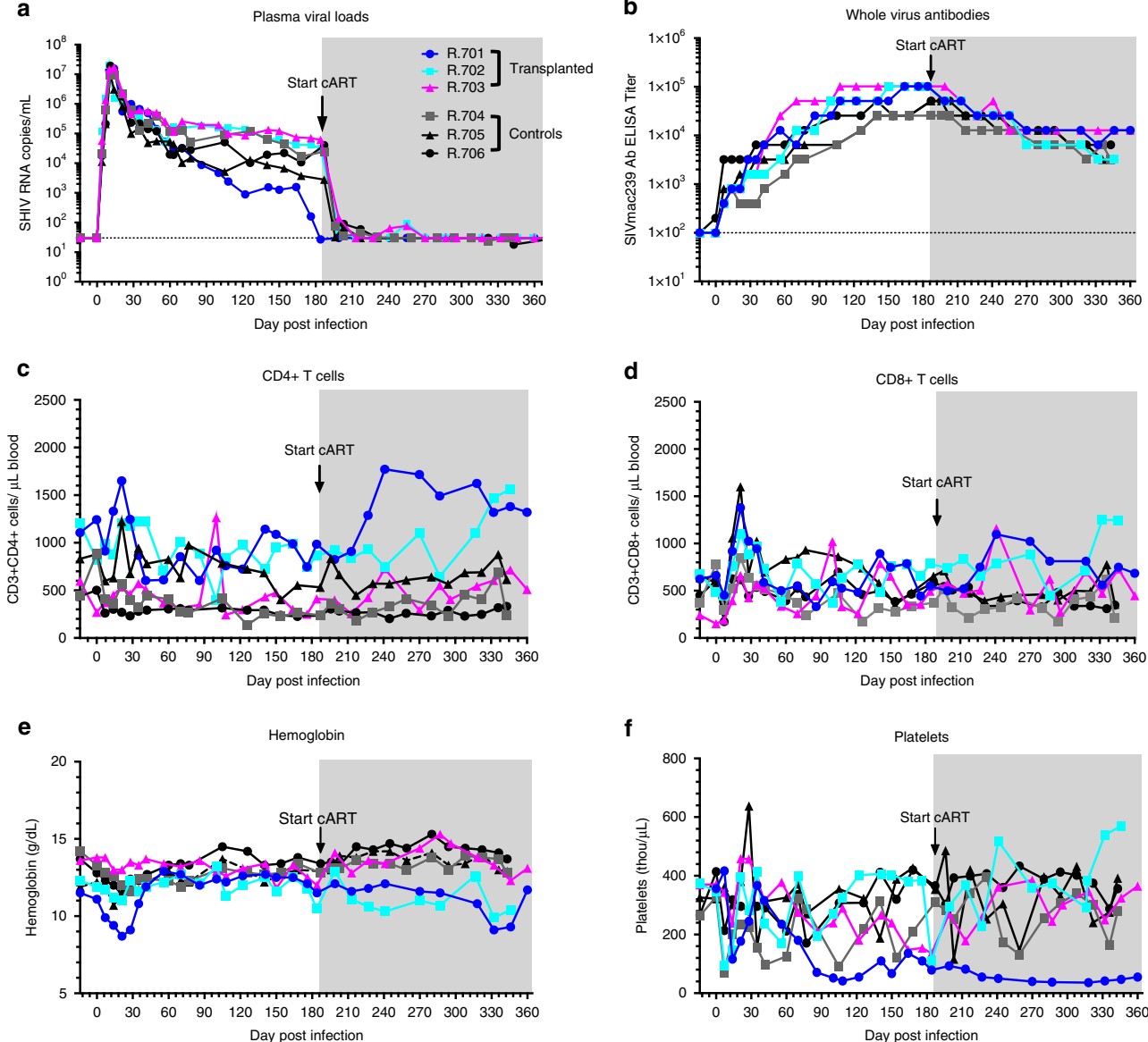

**Fig. 1** Impact of SHIV-C Infection and cART Treatment. Six rhesus macaques were infected intravenously (IV) with SHIV1157ipd3N4 (SHIV-C) and monitored for at least one year prior to allogeneic transplantation. **a** Plasma Viral Loads (PVL) were monitored by quantitative real-time PCR (qPCR) bi-weekly after challenge. cART (PMPA/FTC/Raltegravir) was initiated 6 months post-infection and continued thereafter. The dashed line marks the limit of detection (LOD) of the qPCR = 30 copies/mL. The arrows indicate the day of cART initiation, and the gray shaded areas indicate continuous cART treatment. **b** Whole virus SIVmac239-reactive antibody titers. **c** Peripheral blood CD4+ and **d** CD8 + T cell counts. **e** Hemoglobin (Hgb) concentration (g/dL). **f** Platelet counts (x10$^3$/µL)

animals developed mild anemia (Fig. 1e) and thrombocytopenia (Fig. 1f) during peak viremia, without significant variation thereafter. It should be noted that while five of the six RM studied developed only mild clinical signs of infection, one animal (Animal ID# R.701) developed a clinically significant viral syndrome, including weight loss, lymphadenopathy, and skin rash within 30 days of infection, and displayed a lower peripheral blood viral load compared to the other animals (Fig. 1a). The potential immune reaction to SHIV was accompanied by a significant drop in the platelet count (Fig. 1f). In short, our cohort of animals displayed typical courses of unsuppressed SHIV viremia and responded as expected to cART.

SHIV-C RNA and DNA levels were measured longitudinally in peripheral blood mononuclear cells (PBMC), bone marrow aspirates (BMA), lymph nodes (LN), duodenum, colon, and rectum (Fig. 2). As expected, in each of these tissues, cell-

associated SHIV RNA peaked during untreated infection and declined with cART (Fig. 2a). In contrast, the cell-associated SHIV DNA reservoir persisted in all six tissues even after cART treatment (Fig. 2b), consistent with a large multi-organ viral reservoir in all animals. These results document that significant SHIV reservoirs were present in our animals prior to the allo-HCT intervention.

**Allo-HCT inSHIV+ stably suppressed animals**. Three of the six infected/cART treated animals (R.701, R.702, R.703) were assigned to transplant, based on the availability of an MHC-haploidentical donor (Table 1). The remaining three animals (R.704, R.705, R.706) were observed without transplantation. The transplant strategy employed is illustrated in Fig. 3a. Recipient animals underwent pre-transplant conditioning using total body

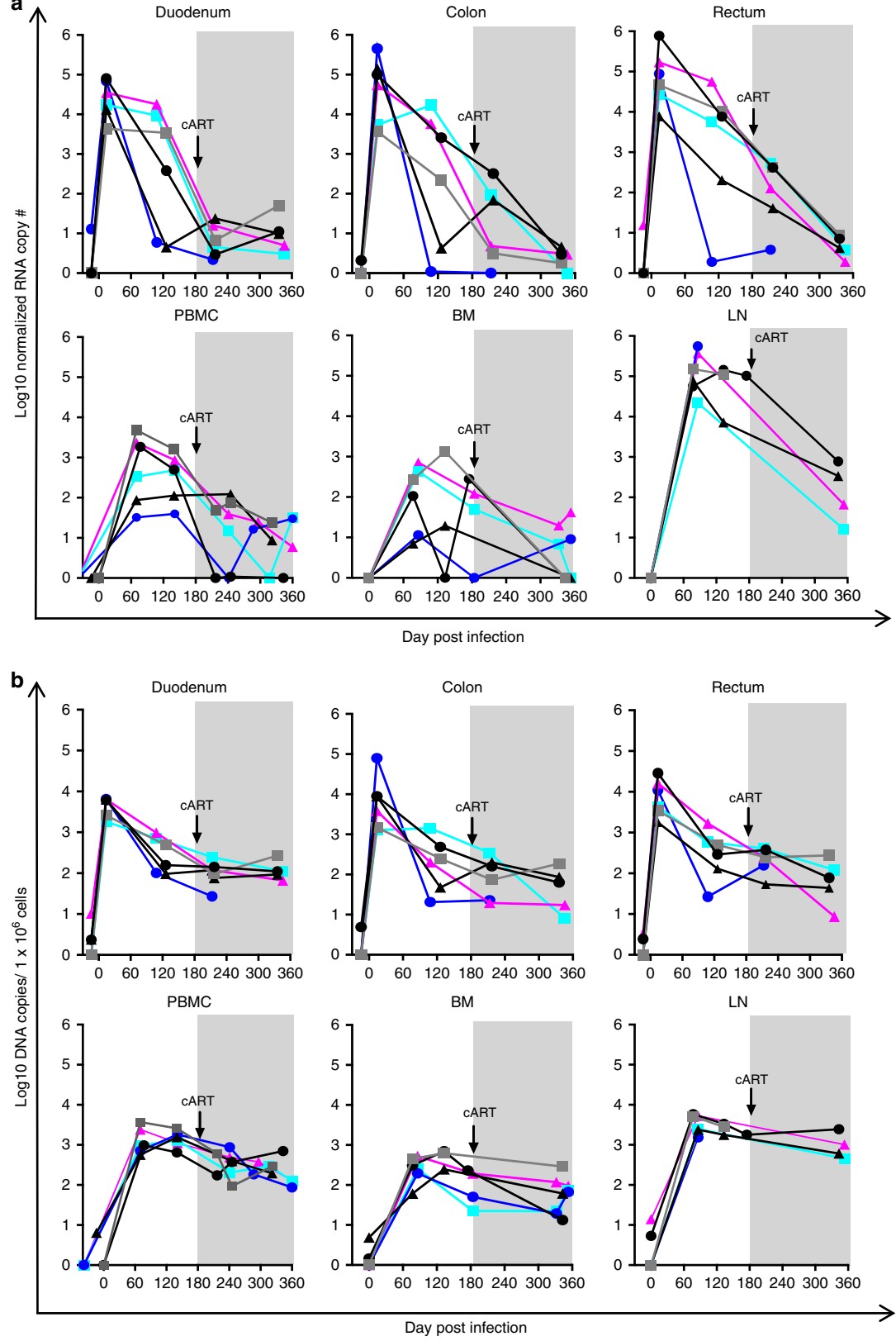

**Fig. 2** Longitudinal reservoir analysis in SHIV-C infected macaques before allogeneic bone marrow transplantation. **a** Six tissues (PBMC, BM, pooled inguinal and axillary LN, duodenum, colon, rectum,) were collected longitudinally at the indicated time points post-infection and cART. Total RNA was extracted from each tissue and subjected to qPCR for viral RNA with GAG specific primers. SHIV genomic RNA was normalized to the crossing threshold value of the genomic standard MRPP30. **b** Total genomic DNA (gDNA) was extracted from the same tissues as in **a**, total SHIV DNA was measured by qPCR, and was normalized to $1 \times 10^6$ cell equivalents with the aid of a genomic DNA standard (macaque RNaseP p30, MRPP30)[35]

**Table 1 Transplant and clinical characteristics**

| Animal ID | R.701 | R.702 | R.703 |
|---|---|---|---|
| TNC/Kg | $5.62 \times 10^8$ | $13.04 \times 10^8$ | $5.71 \times 10^8$ |
| CD34/Kg | $10.5 \times 10^6$ | $13.3 \times 10^6$ | $9.9 \times 10^6$ |
| CD3/Kg | $27 \times 10^6$ | $49.3 \times 10^6$ | $32.5 \times 10^6$ |
| Day of neutrophil engraftment | N/A | 19 | 15 |
| Day of platelets engraftment | N/A | 36 | N/A |
| Survival post transplant | Day + 9 | Day + 47 | Day + 29 |
| Indication for experiment termination | Renal failure | Infection | Acute GVHD |
| Recipient haplotype 1 | A008/B048/DR15a | A023/B012b/DR21c | A023/B043a/DR14 |
| Recipient haplotype 2 | A004/B012/DR02 | A004/B015a/DR22 | A008/B015b/DR13 |
| Donor haplotype 1 | A008/B048/DR15a | A023/B012b/DR21c | A023/B043a/DR14 |
| Donor haplotype 2 | A105/B002/DR-unk | A001/B012a/DR01a | A001/B015b/DR11a |

*TNC* total nucleated cells per kg, *CD34+/kg* CD34 + HSC number per kg, *CD3/kg* total T cell number per kg

irradiation (TBI) as we have previously described (1040 cGy, delivered in 4 divided doses given over 2 days)[17,29]. GVHD prophylaxis included sirolimus (with a target serum level of 5–15 ng/mL), tacrolimus (0.025 mg/kg twice daily, target serum level of 5–10 ng/mL), and mycophenolate mofetil (15 mg/kg twice daily). As shown in Fig. 3a, recipients were transplanted with G-CSF-mobilized bone marrow, which was infused without further gene modification. Total nucleated cell counts (TNC), CD34$^+$ and CD3$^+$ T cell doses, and clinical outcomes for the three allo-HCT recipients are shown in Table 1. Animal R.701, which experienced a significant immune reaction after SHIV-C infection (Fig. 1a) was euthanized at Day 9 post-transplant due to renal failure as confirmed by histopathology. Due to the short time frame of follow up, detailed hematologic reconstitution and donor engraftment measurements were not made in this animal. Animal R.702 was followed for 47 days post-transplant, when terminal analysis was performed due to bacterial sepsis with culture-proven multidrug resistant *Escherichia coli* (Table 1). Animal R.703 was followed for 29 days post-transplant, when terminal analysis was performed due to steroid-resistant acute GVHD (Table 1).

Plasma viral load, hematologic reconstitution, and donor engraftment measurements are shown in Fig. 3b–k. Neutrophil engraftment (defined by absolute neutrophil count (ANC) >$0.5 \times 10^3$/μL for 3 consecutive days) occurred at day 19 and 15 post-transplant, respectively for R.702 and R.703 (Fig. 3d and Table 1). R.702 achieved platelet engraftment (defined as platelet count ≥20,000/μL in the absence of any blood transfusion support for 7 consecutive days) at day 36 post-transplant, while R.703 did not engraft platelets prior to necropsy (Fig. 3f and Table 1). Whole blood donor hematopoietic engraftment was measured by molecular microsatellite analysis as we have previously described[30–33](Fig. 3h), and by flow cytometry using a donor vs. recipient discriminatory Mamu-A01 antibody, which was applicable in both R.702 and R.703 (Fig. 3i–k)[34]. Both animals demonstrated rapid engraftment of donor neutrophils, Fig. 3i), whereas flow cytometric analysis demonstrated distinct patterns of lymphoid engraftment in R.702 and R.703. While R.702 demonstrated near complete CD4+ and CD8 + T cell chimerism in all tissues examined, R.703 demonstrated lower levels of CD4+ and CD8 + T cell chimerism in both the peripheral blood and the organs examined at necropsy (Fig. 3j, k). Collectively, these observations suggest donor engraftment was complete or in progress in both animals at these early post-transplant time points, with high levels of donor myeloid cells, and substantial donor T cells observed in both recipients.

**No evidence of donor anti-SHIV CD8 T cells post-transplant.** One key question that remains unanswered from the Boston Patients' study is whether donor-derived T-cells became infected following transplantation into HIV +, cART-suppressed recipients. In this situation, we would expect to see the development of virus-specific, donor-derived CD8$^+$ T-cells. The MHC-mismatched allogeneic transplant strategy that we employed allowed us to specifically assay for the development of SHIV-specific, donor-derived CD8 + T cells post-HCT in our animals: two of the transplant recipients, (R.702 and R.703) were A01-negative, while donors were A01-positive (Supplementary Table 1). This disparity allowed us to use A01-CM9 tetramers (loaded with the immunodominant SIVmac239-derived Gag peptide CTPYDINQM) to specifically quantify tetramer-positive donor cells post-transplant. As shown in Fig. 4, while a separate cohort of SHIV-infected, cART-treated animals demonstrated measurable tetramer positive CD8+ T cells during the course of infection and cART, we were unable to measure any tetramer-positive cells in recipients who were transplanted with A01+ donors. These data suggest that donor-derived, virus-specific T-cells are not generated following transplantation into SHIV-infected, cART-suppressed recipients.

**SHIV-C reservoir analysis in HCT recipients and controls.** At necropsy, the HCT recipients' SHIV RNA and DNA reservoirs were comprehensively analyzed. For accurate comparison, the three non-transplanted control animals were necropsied at comparable post-cART time-points. The reservoir evaluation comprised 20 tissues, including peripheral blood, bone marrow, lymph nodes (axillary, inguinal, iliac, and mesenteric), duodenum, jejunum, ileum, colon, rectum, spleen, tonsils, thymus, liver, lung, kidney, cerebellum, parietal cortex, hippocampus, and basal ganglia (Figs. 5 and 6). SHIV RNA and DNA measurements were performed using TaqMan real time PCR as previously described[35]. As shown in Figs. 2a and 5, longitudinal and terminal analysis indicated little-to-no SHIV RNA in most organs, with no difference in reservoir size when HCT recipients were compared to untransplanted controls, underscoring the efficacy of cART treatment during HCT in controlling SHIV RNA. Nonetheless, as shown in Fig. 5, low-level SHIV RNA expression was detected in selected sites in individual animals, consistent with prior findings from our group and others[17,36–39].

In contrast to viral RNA measurements, the SHIV DNA reservoir was measurable in all tissues examined, both in untransplanted controls and transplant recipients (Fig. 6). Notably, SHIV-C DNA was detectable even in R.703, which developed GVHD and hence was the best candidate to have

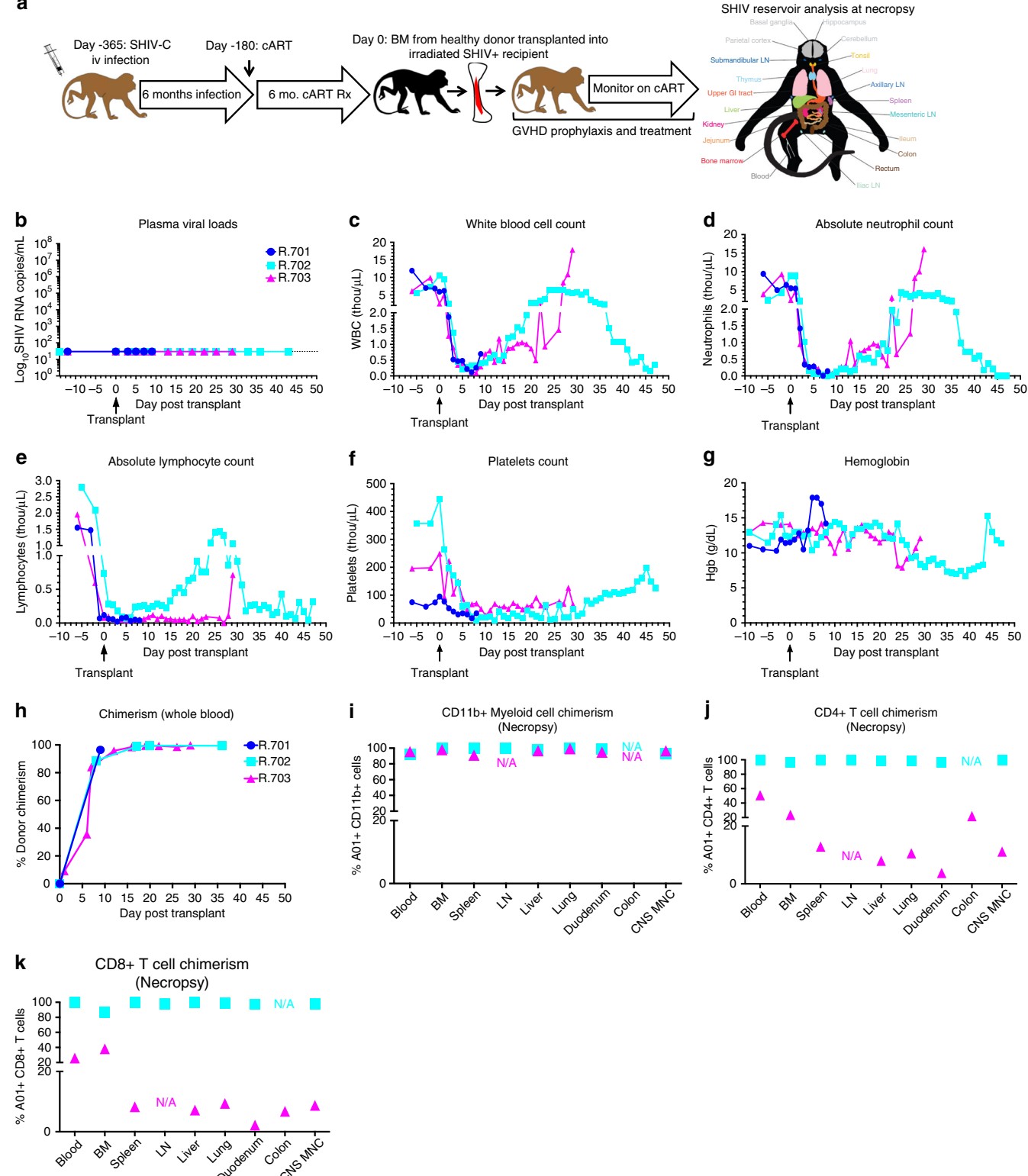

**Fig. 3** Allogeneic HCT in SHIV-C Infected, cART-treated RM. **a** Schematic of SHIV-C infection and allogeneic bone marrow transplantation strategy in SHIV-C infected, cART-treated RM. **b** Plasma Viral Loads (PVL) were measured longitudinally following transplant as specified in Fig. 1a. **c** White blood count (WBC) (×10³/μL). **d** Absolute Neutrophil Count (ANC) (×10³/μL). **e** Absolute Lymphocyte Count (ALC) (×10³/μL). **f** Platelet Count (PLT) (×10³/μL). **g** Hemoglobin (Hgb) concentration (g/dL). **h** Percent whole blood donor chimerism, measured by microsatellite analysis. **i** Percent donor myeloid (CD11b + CD3-) chimerism, measured flow cytometrically. **j** Percent donor CD4 + T cell (CD3+ CD4+ CD8− CD20− CD11b− lymphocytes) chimerism, measured flow cytometrically. **k** Percent donor CD8 + T cell (CD3+ CD8+ CD4− CD20− CD11b− lymphocytes) chimerism, measured by flow cytometry

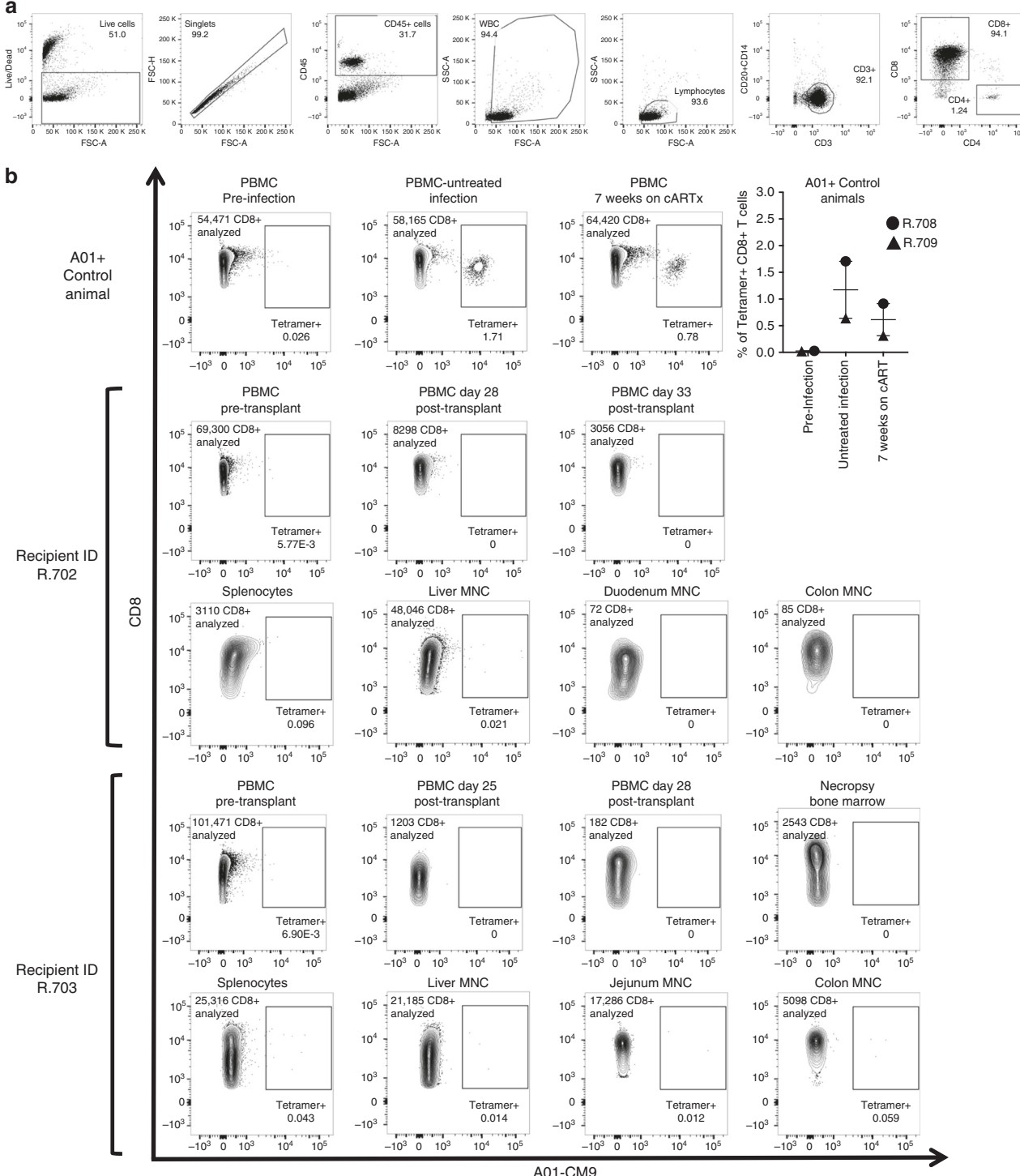

**Fig. 4** Virus-specific donor T-cells are not detected in allogeneic transplant recipients. PBMC and tissue-derived MNC were isolated from four animals: a pair of A01$^+$ animals ('R.708' and 'R.709', with tetramer studies performed pre-infection, at day 90 of untreated infection, and following 7 weeks of cART), and from SHIV+ transplant recipients R.702 and R.703, both A01-negative animals who were transplanted with A01-positive donors. **a** Representative gating strategy from MNC isolated from the jejunum of animal ID R.703. **b** Tetramer + CD8$^+$ T cells were quantified (CD45$^+$CD3$^+$CD8$^+$CD14$^-$CD20$^-$CD4$^-$ lymphocytes). Error bars represent the mean with SEM

experienced a concordant GVVR effect. Importantly, some sites from allo-HCT recipients demonstrated a decrease in SHIV DNA levels compared to pre-transplant samples or untransplanted controls, including PBMC ($P = 0.009$), mesenteric lymph nodes ($P = 0.049$), and spleen ($P = 0.024$) (Fig. 6), consistent with a transplant-dependent reduction in the size of the SHIV reservoir,

as had been observed in HIV+ patients such as the Boston Patients[14,15]. In other tissues, no significant decrease in cell-associated SHIV-C DNA was observed, including bone marrow, pooled inguinal and axillary lymph nodes, iliac lymph nodes, tonsils, thymus, liver, lungs, kidneys, duodenum, jejunum, ileum, colon, and rectum. Comparable levels of the SHIV DNA reservoir

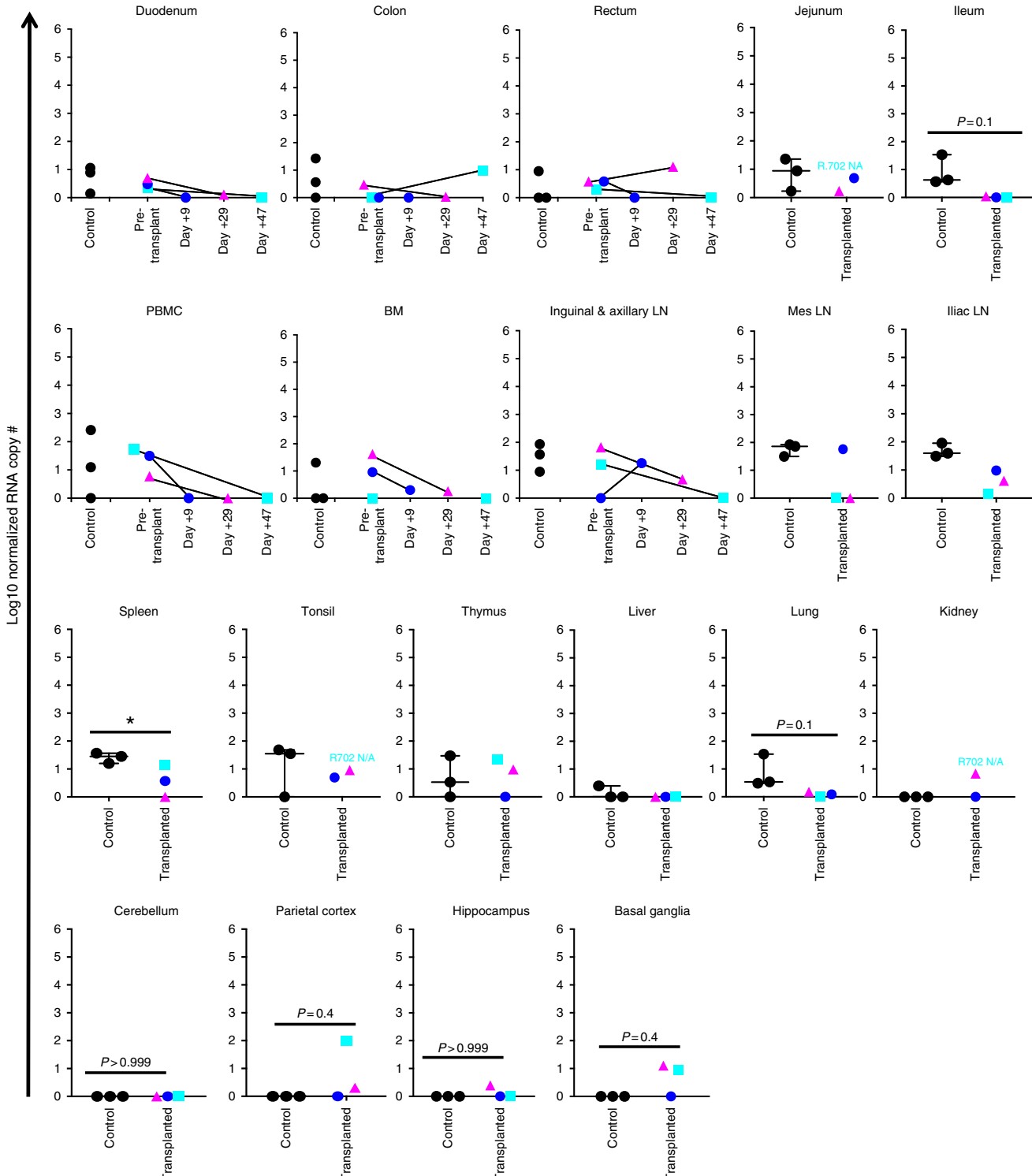

**Fig. 5** SHIV-C RNA in transplanted recipients and controls. Total RNA was extracted from the following tissues at necropsy and subjected to qPCR for viral RNA with GAG-specific primers: PBMC, BM, pooled inguinal and axillary LN, duodenum, colon, rectum, mesenteric LN, iliac LN, spleen, tonsils, thymus, liver, lung, kidneys, jejunum, ileum, cerebellum, parietal cortex, hippocampus, basal ganglia. Where applicable, measurements at multiple time points in transplanted animals are indicated. Error bars represent the mean with SEM. Spleen: $p = 0.0489$, $t = 2.799$, $df = 4$, unpaired two-tailed $t$ test

in transplanted animals and controls were observed even when normalized for the transplant-dependent decrease in CD4+ T-cell counts, which were measurable in bone marrow, pooled inguinal and axillary lymph nodes, iliac lymph nodes, liver, lung, jejunum, and colon (Supplementary Fig. 1). Moreover, for individual animals, there was an increase in CNS-associated SHIV-C DNA levels in the parietal cortex and hippocampus compared to untransplanted controls (Fig. 6). While the t-statistical test has limited power in the setting of group sizes of three animals, these PCR-based reservoir measurements

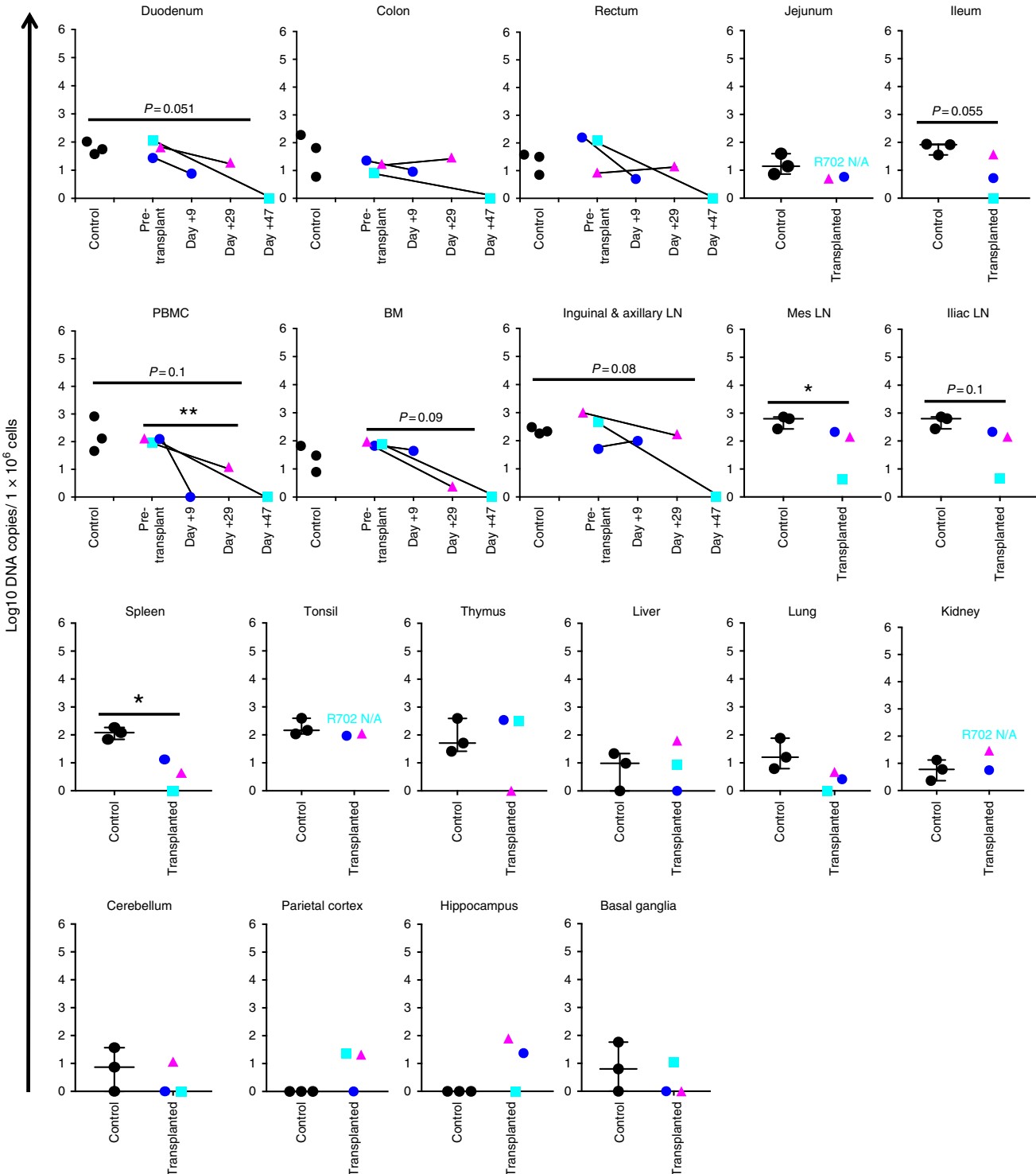

**Fig. 6** SHIV-C DNA in transplanted recipients and controls. Genomic DNA (gDNA) was extracted from the tissues listed in Fig. 5. Total SHIV DNA was measured by qPCR with GAG-specific primers, and normalized to SHIV DNA copy number per $1 \times 10^6$ cells with the aid of a genomic DNA standard (macaque RNaseP p30, MRPP30)[35]. Where applicable, measurements at multiple time points in transplanted animals are indicated. Error bars represent the mean with SEM. *$p \leq 0.05$; **$p \leq 0.01$. Pre-transplant vs. necropsy PBMC: $p = 0.009$, $t = 10.68$, df $= 2$, paired two-tailed $t$ test; mesenteric LN: $p = 0.049$, $t = 2.796$, df $= 4$, unpaired two-tailed $t$ test; spleen: $p = 0.024$, $t = 3.538$, df $= 4$, unpaired two-tailed $t$ test)

demonstrate that cell-associated SHIV-C DNA, and in some cases SHIV RNA$^+$ cells, persist in multiple tissues at early time-points following allo-HCT.

**SHIV DNA measurements in donor vs. recipient subsets**. A key unanswered question in suppressed HIV$^+$ patients undergoing allo-HCT is whether transplanted donor cells can be infected by

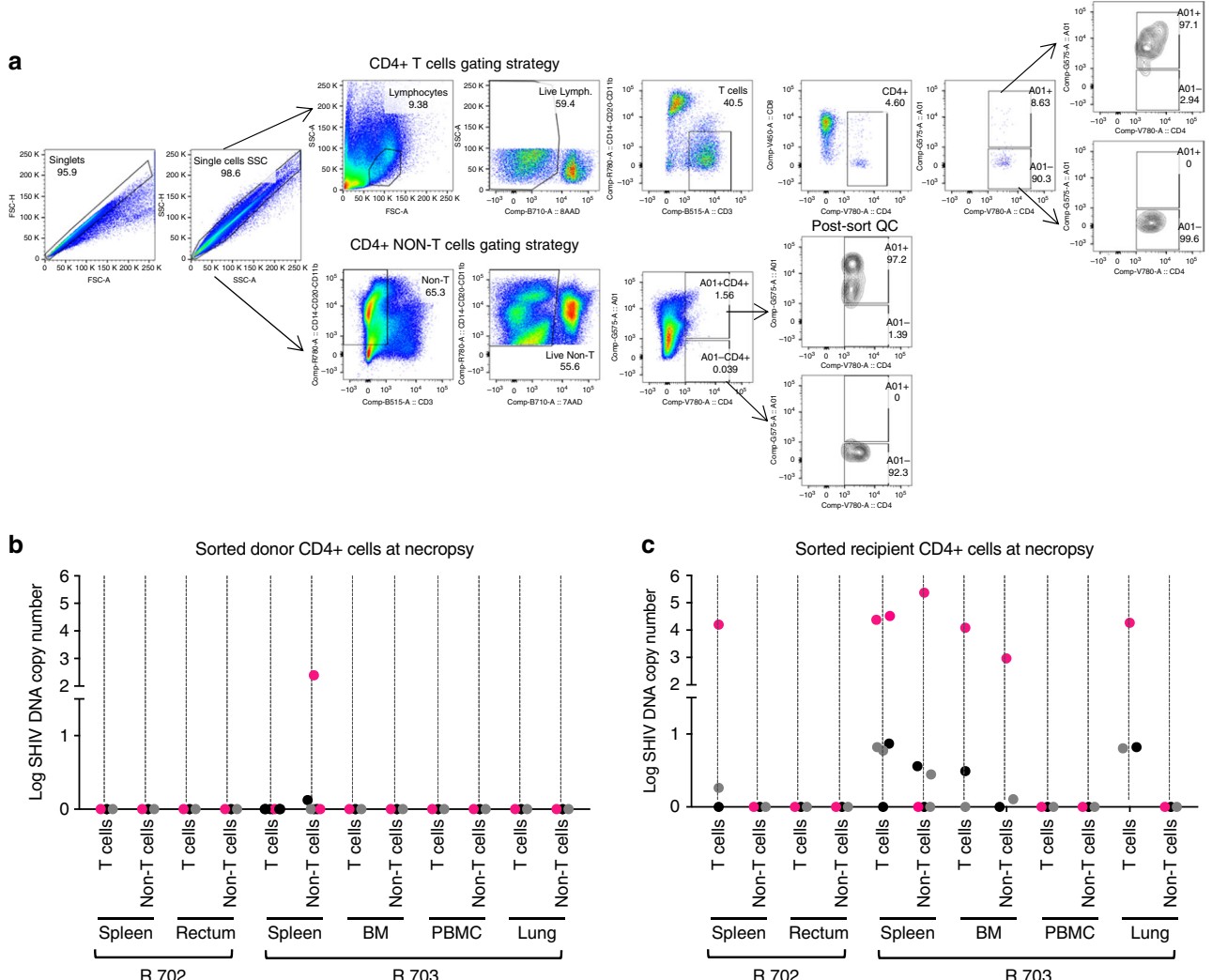

**Fig. 7** SHIV-C DNA levels in sorted donor and recipient CD4$^+$ cells. **a** Gating strategy utilized to sort-purify donor-derived CD4$^+$ T cells (A01$^+$ CD3$^+$ CD4$^+$), donor-derived CD4$^+$ non-T cells (A01$^+$ CD3$^-$ CD4$^+$), recipient-derived CD4$^+$ T cells (A01$^-$ CD3$^+$ CD4$^+$), and recipient-derived CD4$^+$ non-T cells (A01$^-$ CD3$^-$ CD4$^+$). Following the sorting of donor **b** and recipient **c** cells, gDNA was extracted, and SHIV DNA copy number per 10μL of reaction was measured by qPCR with GAG-specific primers in duplicate. The replicate PCR results are shown as gray and black dots (two replicates for each sample with the exception of R.703 spleen, for which two separate samples were analyzed, thus 4 PCR replicates are shown). The raw qPCR results were also averaged and normalized to SHIV DNA copy number per $1 \times 10^6$ CD4 + cells with the aid of a genomic DNA standard (macaque RNaseP p30, MRPP30)[35], which are represented as pink dots

virus-expressing recipient cells, despite ongoing cART. To address this in our NHP model, we leveraged the fact that two of our donor animals carried the Mamu-A01 MHC haplotype, while their transplant recipients, R.702 and R.703, were Mamu-A01 negative. This enabled flow-based sorting for A01-negative (recipient) and A01 + (donor) cells (Fig. 7a), followed by quantification of SHIV DNA in each population by PCR from selected tissues for which sufficient numbers of cells were purified at necropsy (Fig. 7b, c). In recipient R.702, sufficient cells for this assay were available from the spleen and rectum. In recipient R.703, sufficient cells were available from the spleen, bone marrow, peripheral blood and lung. In R.702, we detected SHIV DNA in A01-negative recipient T-cells isolated from the spleen, but not in donor A01 + cells from any samples. In R.703, we detected SHIV DNA in recipient A01-negative cells from the spleen, bone marrow and lungs, in both T cell and non-T cell populations (Fig. 7b, c). In contrast, all A01 + donor cells were negative for SHIV DNA with the exception of a very low level of SHIV DNA

in one of four technical replicates from non-T-cells isolated from R.702's spleen (resulting in an average of $0.25 \times 10^4$ copies per million CD4 + donor cells in this sample compared to an average of $4.82 \times 10^4$ copies of SHIV DNA per million CD4+ measured in recipient cells; Fig. 7b, c). These results suggest that the potential of latently infected recipient cells to transfer virus to transplanted donor cells is extremely limited under cART, and that, early after allo-HCT, the reservoir likely resides in recipient cells. They are also consistent with previous studies that have demonstrated that, when viral DNA is measured in non-T cells, this occurs exclusively in anatomical sites that contain CD4 T cells[40,41].

## Discussion

Three key factors are widely believed to have contributed to the cure of the Berlin patient: (1) the conditioning regimen used for his two transplants, (2) the use of donor cells containing a homozygous HIV-resistant CCR5Δ32 mutation[42], and (3) the

development of post-transplant GVHD[43], which is hypothesized to help clear the HIV reservoir through an as-yet ill-defined GVVR effect. The Boston Patients[14], underwent allogeneic HCT and developed GVHD, but received HIV-susceptible donor cells. Although these patients demonstrated surprisingly long periods of cART-free remission after analytic treatment interruption (ATI), each eventually did experience viral rebound. The data from the Boston patients provided qualitative evidence of HIV persistence despite allo-HCT and GVHD, but could not quantify the sites, recipient-vs.-donor origin, or levels of the residual reservoir after transplant, representing a critical unmet need in the field. In this study, we created a new NHP model to address these key questions. This new model is based on our previously established rhesus macaque allo-HCT model, which is the best-established NHP model of GVHD[29,31,32,44–49]. Our experiments were also strengthened by the fact that rhesus macaques remain the gold standard animal model of HIV infection, prophylaxis, and cure[50]. We believe that these attributes combine to make this the optimal system in which to quantify the impact of allo-HCT on viral reservoir dynamics that were first observed in clinical studies. Our results highlight both the power of the NHP model to exhaustively interrogate viral reservoirs prior to cART interruption, as well as the challenges inherent in these complex, longitudinal studies.

Our initial study plan for these experiments (not shown here) incorporated the most common transplant regimen for patients with haploidentical donors, including non-myeloablative conditioning and post-transplant cyclophosphamide (PT-Cy) as the GVHD prophylaxis platform[51]. However, the first control transplant that was performed with this regimen demonstrated lack of engraftment (Supplementary Fig. 2), a toxicity that has also been described clinically with non-myeloablative HCT and PT-Cy, especially in patients with non-malignant diseases[52–55]. Both in patients and in our animal model, graft failure likely occurred due to a lack of myeloablative pre-transplant conditioning, especially in a transplant recipient not previously treated with anti-leukemia myelosuppressive chemotherapy. To overcome the significant risk of technical failure with non-myeloablative transplant conditioning, and to better model the HIV patients with malignancies that will be the key target population for curative allo-HCT, we adjusted our strategy in SHIV-infected animals to include the myeloablative conditioning regimen and calcineurin inhibitor-based GVHD prophylaxis described here.

This decision also highlights the evolution in thought that has occurred surrounding whether HCT should be considered as a curative therapy for all patients with HIV, particularly for those without associated malignancies. Given the significant potential risks of allo-HCT (including graft rejection, GVHD, and other organ toxicities), and the relative success of lifelong cART, it is now thought that allo-HCT should not be considered as a treatment for HIV patients who do not have other indications for transplant[20]. While this significantly decreases the number of HIV + patients eligible for potentially curative therapy, there are still many HIV + patients who develop malignancies for which HCT is a curative therapy. If those patients could be cured of HIV, two major successes will be achieved: (1) for each individual patient, the cure of HIV would make a tremendous impact on their lives. (2) These transplants could facilitate the acquisition of invaluable data concerning the immunologic requirements for reservoir eradication, which could subsequently be applied in curative treatments that are safer and better indicated for otherwise healthy HIV+ patients.

In building our model of allo-HCT during suppressed SHIV infection, we encountered several challenges. First was the risk of renal insufficiency, a well-described toxicity of cART, most commonly attributed to tenofovir[56,57]. NHP transplant recipients were treated with cART for at least 6 months pre-transplant, and then continued to receive drugs post-HCT. Given the known toxicities associated with allo-HCT (including those associated with irradiation, cyclophosphamide, and calcineurin inhibitors such as tacrolimus) the added renal toxicity of tenofovir likely represented a key limitation in our studies. New formulations of this drug are now available including tenofovir disoproxil fumarate (TDF)[58], which we are incorporating into our current experiments.

The other major issue we encountered with our model was the high risk of infection and GVHD after MHC haploidentical transplant. The Kean laboratory has substantial experience with both MHC-matched and MHC haploidentical transplant in NHP, having established the first MHC-matched NHP allo-HCT model and the only haploidentical NHP HCT model in existence[29,31,32,44–46,48,49]. Given the toxicities we encountered with MHC haploidentical allo-HCT in this study, we are now restricting our studies in SHIV-infected recipients to our previously-reported MHC-identical allo-HCT model. While the risk of GVHD is well-established, the risk of incomplete donor chimerism, in the HIV-infected community deserves special discussion. Given that in this study, we demonstrated the persistence of the SHIV reservoir in recipient tissues, the presence of incomplete donor chimerism would be expected to increase the risk for post-transplant viral rebound, again underscoring the need to develop transplant techniques for HIV-infected patients that yield full donor engraftment. Burwitz et al. also recently reported an allo-HCT model in non-infected animals, using a population of Mauritian-origin cynomolgus macaques (MCM) which express MHC molecules with limited diversity relative to more out-bred species such as rhesus macaques[59]. While the MCM model has several important similarities to our previously-published results[31,32] the data we present in this report argues that the rhesus macaque model likely remains best-suited for HIV GVVR studies for several reasons. First, high donor T-cell chimerism will be essential in order to minimize the number of residual recipient T cells that may harbor replication-competent virus. We are able to generate animals with nearly 100% donor T-cell chimerism without the use of donor lymphocyte infusions, which have only limited relevance to clinical allo-HCT, given their known risks of life-threatening GVHD[60]. Second, although the lack of MHC diversity is an attractive component of the MCM model for feasibility reasons, our previous results make clear that despite their relative MHC complexity, rhesus macaques can be easily and quickly MHC-typed to find MHC-matched pairs, and do not require specialized breeding programs[31,33,61]. Moreover, rhesus macaques are more likely to better represent the diversity of MHC biology in patients than do the genetically bottlenecked MCM population[62]. Finally, the spectrum of clinical outcomes in our rhesus macaque model is quite comparable to those observed by Burwitz et al., facilitating immediate application of the well-characterized rhesus macaque model of HIV infection, as we describe above. In short, our data demonstrate that the well-established rhesus macaque model of allo-HCT is well-placed to interrogate preclinical strategies for HIV+ patients with associated hematological malignancies.

This first study has yielded several important observations that should inform our thinking about the extent of the HIV DNA reservoir and the design of strategies to clear this reservoir. Most striking is the extent of viral DNA in all tissues evaluated, and the lack of complete clearance of the reservoir despite myeloablative irradiation, allogeneic transplantation, and at least in one case, GVHD. As was the case with the Boston Patients, one would expect that, had we performed an ATI in the NHP recipients, viral rebound would have occurred. It is important to note,

however, that follow-up in all three transplant animals was short, and thus, the possibility certainly exists that further reductions in the recipient reservoir may have occurred with longer follow-up. Whether these long-term reductions would have been sufficient to clear the reservoir is, of course, an open question. Importantly, while it was impossible with the Boston patients to identify an occult reservoir prior to ATI, the NHP model allowed us to quantify the potential sources of this reservoir and its donor-vs.-recipient origin. As in the Boston patients, this reservoir did not reside in the peripheral blood, underscoring the challenge in adequately evaluating clearance of the HIV reservoir in patients, in whom extensive tissue sampling is not feasible. The second important observation from this study was the fact that, while cell-associated SHIV-C DNA was at or below the limit of detection in the periphery (i.e. PBMC) and in several lymphoid and non-lymphoid tissues (including the tonsils, thymus, liver, and kidneys), the reservoir was not decreased after allo-HCT in other tissues, and moreover, in both the parietal cortex and the hippocampus, the viral DNA reservoir was higher in some of the transplant recipients compared to untransplanted controls. Although total viral DNA measurements have been demonstrated to overestimate the size of the replication-competent viral reservoir[63], this observation raises the possibility that transplant recipients, at least early after HCT, may experience an increase in viral DNA in some tissue sites, perhaps due to impaired anti-viral immunity early after transplant[17]. Importantly, while we detected SHIV DNA in residual recipient-derived cells in many different tissues, we only observed (very low-level) cell-associated viral DNA in donor cells in a single replicate (of four) in one tissue, from a non-T-cell (CD3⁻negative) CD4+ population from one of the animals analyzed. This could be true infection of a donor cell, or phagocytosis of a dying, infected recipient cell[40,64]. This finding suggests that early after transplant, infection of donor cells occurs at extremely low levels, if at all. While it remains an open question whether there could be late transfer of virus to donor cells, these results imply that long-lived recipient cells will represent the key viral reservoir after transplant. The third critical observation concerns the lack of evidence for the development of donor-derived anti-SHIV CD8 + T cells post-transplant. This may have been caused by two phenomena: First, donor cells were exposed to post-transplant immunosuppression, which would be expected to blunt the development of anti-HIV immunity. Second, as noted above, the post-transplant SHIV reservoir resided in recipient, and not donor cells, which is consistent with the lack of development of anti-viral immunity in donor cells post-transplant. These results support our interpretation that, in the presence of anti-retroviral therapy, the ability of donor cells to evolve an anti-SHIV CD8 + response will be very limited after transplant. The observation of measurable viral DNA in the brain and other organs of these transplant recipients in the setting of a lack of evidence for donor-derived anti-SHIV immunity is a sobering reminder of the extent and tenacity of the HIV reservoir. Fully eradicating this reservoir, even with the ultimate latency reversing[41] maneuver of myeloablative TBI and allogeneic HCT, will likely require further combinatorial interventions, likely including gene modification of donor hematopoietic stem and progenitor cells (HSPCs) to resist infection and target infected cells.

In summary, we report the first NHP model of allo-HCT in SHIV-infected, cART treated NHP. This work establishes both the feasibility as well as the challenges inherent in developing these complex but highly clinically relevant models, and the significant added-value of NHP models compared to patient studies in the ability to achieve detailed, whole-organism interrogations of the viral reservoir both before and after transplant. We have documented that the wide-spread,

multi-organ SHIV DNA reservoir persists despite allo-HCT. Our work suggests that allo-HCT for HIV should focus on the lowest risk transplant paradigms, given the potentially greater risks of toxicity in these patients, and that optimizing the anti-viral reservoir effects, for example through the genetic incorporation of HIV resistance factors, remains a critical area for future investigations.

## Methods

**NHP ethics statement**. This study was conducted at the Washington National Primate Research Center (WaNPRC), an AAALAC accredited program, in accordance with the regulations detailed in the U.S. Department of Agriculture Animal Welfare Act and in the Guide for the Care and Use of Laboratory Animals of the National Institutes of Health. It was approved by University of Washington Institutional Animal Care and Use Committee.

**SHIV challenge and infection monitoring**. Animals were infected IV with 9500 TCID$_{50}$ SHIV-1157ipd3N4 (SHIV-C)[23,35,65]. cART treatment comprised the following agents: tenofovir ('PMPA') (20 mg/kg, given once daily subcutaneously), emtricitabine ('FTC') (40 mg/kg, given once daily subcutaneously) and raltegravir (150 mg per dose, given orally twice daily)[23]. PMPA and FTC were a kind gift from Gilead Sciences, Inc., and raltegravir was a kind gift of Merck. Plasma viral load measurements[24] and quantification of whole virus reactive antibodies[66] were carried out as previously described. Briefly, viral RNA was isolated from plasma collected from blood drawn into EDTA. Reverse-transcribed and unspliced SHIV RNA was measured by TaqMan real time PCR with the following primers: GAG5f (5′-ACTTTCGGTCTTAGCTCCATTAGTG-3′), GAG3r (5′-TTTTGCTTCCTCA GTGTGTTTCA-3′), and the TaqMan probe GAG1tq (FAM-TTCTCTTCTGCGT GAATGCACCAGATGA-TAMRA)[24]. The limit of detection of our plasma viral load RT-PCR assay was 30 copies/mL. Whole virus SIVmac239-reactive antibodies titers were measured by ELISA as previously described[66]. The antibody titers were calculated as the reciprocal of the highest serum dilution that resulted in an optical density reading greater than the average values obtained with negative macaque sera plus three standard deviations[35,66]. Complete blood counts and CD4/CD8 immunophenotyping were carried out as previously described[65].

**NHP transplant study design**. Six RM were infected with SHIV-C[25] (kindly provided by Dr. Ruth Ruprecht, Texas National Biomedical Research Center) for 6 months, following by cART (PMPA/FTC/raltegravir) for a minimum of 6 months. At this time, animals were divided into two cohorts: (1) Three animals (R.701, R.702, and R7.03), received an allogeneic bone marrow transplant from an uninfected donor and (2) Three animals (R.704, R.705, and R.706) were left untransplanted. We used both microsatellite and allele specific MHC typing[31] to perform MHC typing on the donor and recipient animals (Table 1). All transplant recipients were half siblings and MHC haploidentical with their respective donors. During each of the experimental phases (pre-infection, untreated infection, cART-treated infection, and cART-treated transplant), we sampled six tissues longitudinally (blood, bone marrow, axillary/inguinal LN, duodenum, colon, and rectum) to monitor SHIV RNA and DNA over time. Upon euthanasia, we collected 14 additional tissues (mesenteric LN, iliac LN, spleen, tonsils, thymus, liver, lung, kidneys, jejunum, ileum, cerebellum, parietal cortex, hippocampus, basal ganglia) for terminal reservoir analysis.

**Transplant strategy**. We modified our previously established transplant strategy[32] to the SHIV-infected cART-treated RM used in this study. G-CSF-mobilized bone marrow was used as a stem cell source, with donors given G-CSF (Amgen) at 50 μg/kg for three days prior to bone marrow harvest. Bone marrow was infused into recipients without further manipulation. Table 1 shows the cell doses obtained for each recipient. Recipients received pre-transplant conditioning with a myeloablative dose of TBI (1040 cGy, divided in 4 doses over 2 days) delivered via a Varian Clinac 23EX Energy Linear Accelerator (Varian), at a dose rate of 7 cGy/min. Transplant recipients received a central venous catheter surgically placed on the day of transplant. In addition to cART, transplant recipients received antiviral and antibacterial prophylaxis with acyclovir (5–10 mg/kg IV daily), cidofovir (3–5 mg/kg IV weekly), vancomycin (5–10 mg/kg daily), ceftazidime (150 mg/kg IV daily), and fluconazole (5 mg/kg orally or IV daily) as previously described[29]. We administered blood support based upon clinical need (i.e. when platelets counts decreased below $50 \times 10^3$ per μL, or hemoglobin was below 9 g/dL, or significant hemorrhage was noted). Blood support comprised irradiated whole blood or platelet-rich plasma (PRP), following passage through a LRF10 leukoreduction filter (Pall Medical), in adherence to the ABO typing. GVHD was evaluated as previously described[32]. When present, GVHD was treated with corticosteroids. Transplant recipients were euthanized in consultation with WaNPRC veterinarians and in accordance with humane endpoint guidelines, and control animals were euthanized to closely match the days post infection and post cART of euthanasia of each transplant recipient.

**Chimerism analysis.** Donor chimerism was monitored by molecular chimerism analysis-based divergent microsatellite markers as previously described[30,31], as well as by flow cytometry using a discriminatory anti-Mamu A01 antibody[34]. Flow cytometric analysis was performed with FlowJo (Tree Star) using the following analysis pipeline: (1) Doublets were excluded on the basis of FSC-H vs. FSC-A and SSC-H vs. SSC-A gating, (2) Neutrophils cells were identified as high SSC, CD11b + CD3- cells, (3) T cells as CD3 + CD20- CD11b- cells present in the lymphocyte gate (then sub-gated for CD4 or CD8 positivity), and (4) B cells as CD3- CD20 + lymphocytes. The following antibody clones were used for this analysis: anti-Mamu A01 clone P12[34] (Anti-Mamu A01-PE, NHP Reagent Resources,1:50 dilution), anti-CD11b clones ICRF44 and D12 (BD Biosciences, catalog numbers 557918 and 340936, 1:100 dilution), anti-CD3 clone SP34–2 (BD Biosciences, catalog number 557757, 1:50 dilution), anti-CD4 clone OK-T4 (Biolegend, catalog number 317442, 1:100), anti-CD8 clone RPA-T8 (BD Biosciences, catalog number 563795, 1:50 dilution), anti-CD20 clone 2H7 (ThermoFisher, catalog number 45-0209-42, 1:50 dilution).

**Tetramer staining experiments.** PBMC and tissue-derived MNC were thawed and subjected to Live/Dead Aqua fluorescent reactive dye (ThermoFisher) in PBS for 30 mint at RT. The cells were washed cells with RPMI + 10% FBS and subsequently incubated with 3µg/mL A01-CM9 tetramers (NIH Tetramer Core Facility and MBL International) in PBS + 2% FBS at room temperature for 35 min at $2 \times 10^6$ cells/100 µL in polypropylene FACS tubes. The cells were then incubated at 4 °C with antibodies directed to cell surface markers for additional 30 min. After 2 washes with PBS + 2% FBS, the cells were fixed with 200 µL of stabilizing fixative (BD Biosciences) at room temperature. Tetramer + CD8 + T cells were quantified with the aid of a Fortessa flow cytometer (BD) by gating on live, singlets, CD45 + , CD8 + CD3 + CD14- CD20- CD4- cells present in the lymphocyte gate (CD45-PerCP, BD Biosciences, catalog number 558411, 1:10 dilution; CD8-BUV395, BD Biosciences, catalog number 563795, 1:50 dilution; CD3-FITC BD Biosciences, catalog number 556611, 1:5 dilution; CD14-PE-Cy7, BD Biosciences, catalog number 557742, 1:100 dilution; CD20-PE-Cy7, BD Biosciences, catalog number 560735, 1:50 dilution; CD4-BV785, Biolegend, catalog number 317422, 1:100 dilution).

**Analysis of cell associated SHIV RNA and DNA from tissues.** At the time of necropsy, animals were perfused with 0.5 liter of PBS/kg immediately following euthanasia in order to minimize blood contamination of tissues. ~5 mm$^3$ fragments isolated from 20 tissues were preserved in RNAlater (Thermo Fisher) overnight at 4 $^0$C, and the morning after, were blotted dry, snap frozen, and transferred to −80 $^0$C. Tissue fragments were homogenized in RLT Plus lysis buffer (Qiagen) with the aid of a Precellys 24 Homogenizer and CK28-R hard tissue homogenizing beads (Bertin Corp)[17]. Total RNA and gDNA were isolated from tissue homogenates with the aid of RNA Plus mini kit and QiAmp DNA blood mini kit, respectively (Qiagen). SHIV DNA was quantified by TaqMan real time PCR with the aid of GAG-specific primers and a SIVGAG probe, specifically: SIVGAG Probe [6 FAM] TGTCCACCTGCCATTAAGCCCGA [BHQ1], SIVGAGF GCA GAG GAG GAA ATT ACC CAG TAC, SIVGAGR CAA TTT TAC CCA GGC ATT TAA TGT T[67]. Cellular genome equivalents were measured using both primers and a probe specific for the macaque ribonuclease P protein subunit p30 (RPP30) gene, specifically: MRPP30 Probe [VIC] TGTGACCTGAAGGCTCTGCGCG [BHQ1], MRPP30F GACTTGGACGTGCGAGCG, MRPP30R GCCGCTGTCTCCACAAGT. For viral RNA quantification, cDNA was reverse-transcribed from total RNA and then subjected to nested-PCR in an end point dilution series[35]. Normalized SHIV RNA copy number in tissues was calculated by normalizing SHIV RNA copy number to the crossing threshold of macaque RNase P subunit p30 RNA[17].

**Cell sorting experiments of recipient vs. donor cells.** Cells were obtained for flow cytometric sorting as follows: tissue resident mononuclear cells (MNC) were isolated from lung and rectum tissue by first mincing the organs over a cutting board, followed by respective collagenase IV (Life Technologies) or liberase TL (0.26 U/mL, Roche) digestion in HBSS buffer for 1 hour, with continuous agitation at 37 °C in the presence of DNAse I (0.2 U/mL; Roche). The digested lung and rectum tissues were forced through a metal strainer, then sequentially passed through 100 µm, 60 µm, and 40 µm nylon filters. The resulting cell suspensions were mixed 1:3 with a 90% Percoll solution (VWR), and overlayed over 10 mL of a 70% Percoll solution, following by centrifugation at 800 × g for 30 min at room temperature with no brake. PBMC were purified over a ficoll gradient (Ficoll-Paque Plus, GE Healthcare) by centrifuging at 400 × g for 20 min at room temperature with no brake. Spleen tissue was dissociated through a metal strainer then subjected to red blood cells (RBC) lysis followed by two washes with PBS. Bone marrow cells were isolated from femurs and filtered over a 70 µm nylon filter followed by RBC lysis and two washes with PBS. The purified cells were labeled with the following antibodies and flow cytometrically sorted using a FACSARIA III (BD) using the following gating strategy (Fig. 7a): (1) Doublets were excluded on the basis of FSC-H vs. FSC-A and SSC-H vs. SSC-A gating; (2) live cells were selected as 7AAD- cells; (3) CD4 + T cells were sorted as CD3 + CD4 + CD8- CD20- CD14- cells present in the lymphocyte gate, then sub-gated as A01 + donor vs. A01- recipient T cells; (4) CD4 + non-T cells were sorted as CD3- Lin + (positive in the CD20/

CD14 channel), then were sub-gated as CD4 + A01 + donor vs. CD4 + A01- recipient non-T cells. The following antibody clones were used: anti-Mamu A01 clone P12[34] (Anti-Mamu A01-PE, NHP Reagent Resources,1:50 dilution), anti-CD3 clone SP34-2 (BD Biosciences, catalog number 557757, 1:50 dilution), anti-CD4 clone OK-T4 (Biolegend, catalog number 317442, 1:100), anti-CD8 clone RPA-T8 (BD Biosciences, catalog number 563795, 1:50 dilution), anti-CD14 clone M5E2 (BD Biosciences, catalog number 557742, 1:100 dilution), and anti-CD20 clone 2H7 (BD Biosciences, catalog number 560735, 1:50 dilution). In samples where enough cells were available post-sort, we performed a quality control analysis to assess the purity of the A01 + and A01- cells, with most samples tested showing >90% purity (Supplementary Table 1).

**Statistical analysis.** Values within all groups were first checked for Gaussian distribution using the Shapiro-Wilk normality test. Each symbol represents a single study animal. Values are presented as mean±standard error of the mean (SEM). For SHIV DNA and RNA comparisons between control and transplanted animals, and between pre- and post-transplant values within the same animal, we utilized the two-tailed unpaired and the paired Student's $t$ test for normally distributed values, respectively, or the Wilcoxon test and the Mann–Whitney test for non-normally distributed data. To calculate statistical significance between repeated measures (i.e. CD4+ and CD8+ T cell numbers, platelets numbers, and hemoglobin concentrations over time), we employed the one way analysis of variance (ANOVA) for normally distributed values, and the Friedman test for non-normally distributed values, both followed by the Bonferroni correction to adjust for the number of comparisons made. Comparisons were considered statistically significant at $p \leq 0.05$. All tests were carried out with the aid of Prism software (GraphPad).

## Data availability

The authors declare that all the data supporting the findings of this study are available within the paper and its supplementary information files, or from the corresponding author upon request.

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

## Acknowledgements

We thank Ruth Ruprecht for the SHIV-C inoculum, Gilead and Merck for cART drugs and the Virology Core personnel, Bing Mei and Baoping Tian at the WaNPCR for assistance with viral load and serology assays. We thank the NIH Tetramer Core Facility for the kind provision of A01-CM9 tetramers. This study was supported by grants from the National Institutes of Health, National Institute of Allergy and Infectious Diseases (U19 AI096111 and UM1 AI126623 to HPK and KRJ, U19-AI051731, AI116184, and UM1AI126617 to LSK), National Heart, Lung, and Blood Institute (R01 HL116217 and U19 HL129902 to HPK and LSK and R01HL095791 to LSK). HPK is a Markey Molecular Medicine Investigator and received support as the inaugural recipient of the José Carreras/E. Donnall Thomas Endowed Chair for Cancer Research and the Fred Hutch Endowed Chair for Cell and Gene Therapy.

## Author contributions

L.C. and C.W.P. coordinated the project, performed the experiments, analyzed the data, assembled the figures, and wrote the manuscript. JBS coordinated the project and performed experiments. V.T. and S.N.F. performed experiments and edited the manuscript. J.M.C., M.B., A.Y., S.R., W.M.O., P.S.P., H.M., H.Z., A.T., D.J.H., K.B., L.S., M.L.W.H. performed experiments. K.Z., M.H., J.O., V.N., C.A.A., J.F.L., K.V., C.E.H., C.E., S.W., B. A., J.A., N.I., A.M., and C.M. provided veterinary support. A.B. and R.D.M. performed pathologic analysis. S.H. directed the U.W. viral core performing virological assays and edited the manuscript. K.R.J. directed the D.N.A. and R.N.A. quantification experiments and edited the manuscript. H.P.K. and L.S.K. conceived, designed, and directed the project and wrote the manuscript.

## Additional information

**Competing interests:** The authors declare no competing interests.

