## [Peer Review File · Nature Communications]

Reviewers' comments:

Reviewer #1 (Remarks to the Author):

Colonna et al. address the question whether allo-HCT can eradicate the SHIV reservoir. While a full donor chimerism was achieved the animals did not clear the virus in different organs. The result is not unexpected but described for the first time in NHP the tissue localization. The experiments are well performed and the conclusions are supported by the data. The question is how much this report advances the field.

Major comments:

1. Figure 4: How can one perform a reliable statistic test with such a small number of animals?
2. The data are mainly descriptive. Have the authors attempted to perform IFN gamma release assays with the allogeneic T cells to test if there are virus specific T cells?
3. Have tetramer stainings for virus specific T cells been performed?
4. The authors base their strategy on the report of one HIV patient who was cured by allogeneic SCT. However it is important that the donor of this patient was CCR5 deficient and the HIV was CCR5 dependent. In contrast to this situation the authors used MHC-haploidentical allogeneic donors during continuous cART.

Reviewer #2 (Remarks to the Author):

Comments to authors: Here the authors have used their model of myeloablative chemotherapy and total body irradiation followed by allogeneic stem cell transplant of previously SIV-infected and ARV-treated NHPs. Following allogeneic stem cell transplant the animals were sacrificed and analysis of viral genetic material was assessed by immunohistochemistry. The authors find that their procedure did not cure the SIV-infected animals and that the remaining virally-infected cells were all recipient. These data suggest that even very aggressive myeloablative/HSCT protocols will not, routinely, lead to a cure for HIV+ individuals as infected recipient cells remain. These findings are consistent with recent small scale human trials. Overall the authors provide compelling information that they have created a reliable model for HSCT which can be used in SIV+ NHPs and that this model can be used to aggressively assess SIV target cell maintenance.

1. The largest, and most obvious, issue with the manuscript is the animal number. Of the animals who were transplanted, we're really talking about one animal (R.702). 701 died early and 703 was inefficiently transplanted (poor T cell chimerism). In many tissues 702 had the lowest level of viral DNA and RNA, thus there may be hope that such an approach might actually "work". Given the incredible effort that these studies require, it is reasonable that such a small cohort was used. However, the authors really should temper their

discussion. The authors should discuss and highlight the complications with this type of approach for "cure", including GVHD and incomplete chimerism.

2. The sorting analysis shown in Figure 6 is concerning and some in the field are going to scoff. From the FSC/SSC the authors missed some leukocytes. For these types of experiments CD45 is definitely a preferred method to identify leukocytes. The authors are strongly encouraged to use CD45 in the future.

3. In figure 6 the authors should color code the text of the animals' names so that they match the other figures (red and green).

4. The authors only find viral DNA in "non T cells" in anatomical sites that contain CD4 T cells. This is consistent with the data found in references 62 and 63. The authors should comment on this.

5. The authors find equal, or better, chimerism among myeloid cells. This is not consistent with myeloid cells having longer half lives compared to T cells. Indeed, that myeloid cells are long lived is extremely dogmatic and accumulating data are challenging this finding. The authors should provide discussion as to how their chimerism data suggest myeloid cells turn over just as much as lymphocytes.

Editor's comments:

...We therefore invite you to revise and resubmit your manuscript, taking into account the points raised. In particular, additional experiments testing for virus specific T cells along the lines suggested by reviewer 1 would strengthen the case for publication with us....

Based on the reviewer's suggestion, we have now performed tetramer assays to quantify virus-specific T cells in transplant recipients. This experiment demonstrates one of the major advantages of the transplant strategy that we describe: the ability to specifically track donor engraftment, and now, donor anti-SHIV immunity, via unique MHC mismatches between donor and recipient. As shown in **Figure 4** of the revised manuscript, we used anti-CM9 tetramers (loaded with the immunodominant SIVmac239-derived Gag peptide CTPYDINQM) and distinguished donor/recipient pairs on the basis of *Mamu-A*01* status. While A^*01^+ Tetramer⁺ cells could be enumerated both before and after ART treatment in A^*01^+ , SHIV-infected animals, virus-specific donor cells were not detectable after transplant into A^*01^- recipients.

This finding is likely the result of the transplant-related immunosuppression from the TBI and post-transplant immunosuppressive therapy used to control GVHD, which would be expected to blunt the development of anti-HIV immunity. In addition, the post-transplant SHIV reservoir resided in recipient cells and not in donor cells, which is likely due to the continuous cART treatment during the transplant. We believe that this lack of new seeding of the donor reservoir also contributed to the absence of any anti-viral immunity in donor, A^*01^+ cells post-transplant. These results support our interpretation that in the presence of anti-retroviral therapy the ability of donor cells to evolve an anti-SHIV CD8+ response will be very limited after transplant. A discussion of these new results is found on **Pages 2, 8-9 and 16** of the revised manuscript.

Reviewer's Comments:

Reviewer #1:

Major comments:

1. Figure 4: How can one perform a reliable statistic test with such a small number of animals?

While it is possible to determine a t-statistic when comparing $n = 3$ for two groups (especially in the setting of paired samples), we appreciate the reviewers point that the statistical power is marginal with these small numbers. We have added a statement to the text that underscores this point on **Page 10** of the revised manuscript.

2. The data are mainly descriptive. Have the authors attempted to perform IFN gamma release assays with the allogeneic T cells to test if there are virus specific T cells? While the post-transplant cryopreserved cells were not able to be successfully stimulated for IFN γ release assays, we were able to perform tetramer assays (described in detail below). We believe that these tetramer assays provide critical new data linking the lack of a donor reservoir with the absence of a donor-T cell anti-SHIV immune response post-transplant (shown in **Figure 4** of the revised manuscript).

3. Have tetramer stainings for virus specific T cells been performed?

(Please see our answer to the editor's comments, above, which detail our newly-performed tetramer staining experiments).

4. The authors base their strategy on the report of one HIV patient who was cured by allogeneic SCT. However it is important that the donor of this patient was CCR5 deficient and the HIV was CCR5 dependent. In contrast to this situation the authors used MHC-haploidentical allogeneic donors during continuous cART.

The goal of our study was to develop a rigorous system in which the components of the Berlin Patient's cure could be systematically tested in a NHP system. The current study builds on our previous work (Mavingner M et al., PLOS Pathogens 2014 and Peterson CW et al., JCI Insight 2017) that demonstrated that pre-transplant conditioning alone is insufficient to clear the reservoir after autologous transplantation. In the current manuscript we present the first allo-HCT model of transplant mediated reservoir cure, and have focused on wildtype (not CCR5 deficient) cells to determine the impact of allo-HCT alone on the reservoir. Our results provide an important rationale for further studies employing allo-HCT along with HIV resistance factors (including CCR5 deletion).

Reviewer #2:

1. The largest, and most obvious, issue with the manuscript is the animal number. Of the animals who were transplanted, we're really talking about one animal (R.702). 701 died early and 703 was inefficiently transplanted (poor T cell chimerism). In many tissues 702 had the lowest level of viral DNA and RNA, thus there may be hope that such an approach might actually "work". Given the incredible effort that these studies require, it is reasonable that such a small cohort was used. However, the authors really should temper their discussion. The authors should discuss and highlight the complications with this type of approach for "cure", including GVHD and incomplete chimerism.

We thank the reviewer for this helpful comment, and have now tempered our discussion of the impact of allo-HCT on the SHIV reservoir (this is now found on **Page 15** of the revised manuscript). We have also expanded our discussion of the complications associated with a transplant-based cure, including GVHD and incomplete chimerism. This enhanced discussion is found on **Page 14** of the revised manuscript.

2. The sorting analysis shown in Figure 6 is concerning and some in the field are going to scoff. From the FSC/SSC the authors missed some leukocytes. For these types of experiments CD45 is definitely a preferred method to identify leukocytes. The authors are strongly encouraged to use CD45 in the future.

We thank the reviewer for this important comment. When we designed the original experiment, our reasoning was that given that most of the mononuclear cells analysed were derived from hematopoietic organs (i.e. blood, bone marrow, spleen), an anti-CD45 antibody wasn't required. However, we agree that the addition of this antibody would have been optimal, and the new tetramer experiments that we have now performed (**Figure 4** of the revised manuscript) included an anti-CD45 antibody.

3. In figure 6 the authors should color code the text of the animals' names so that they match the other figures (red and green).

We have now made this change.

4. The authors only find viral DNA in “non T cells” in anatomical sites that contain CD4 T cells. This is consistent with the data found in references 62 and 63. The authors should comment on this.

We have now added a discussion of this finding on **Page 11** of the revised manuscript.

5. The authors find equal, or better, chimerism among myeloid cells. This is not consistent with myeloid cells having longer half-lives compared to T cells. Indeed, that myeloid cells are long lived is extremely dogmatic and accumulating data are challenging this finding. The authors should provide discussion as to how their chimerism data suggest myeloid cells turn over just as much as lymphocytes.

The myeloid cell chimerism we report is predominated by neutrophils, which have a very short half-life. We have now clarified this in the manuscript (**Page 8, 20**) of the revised manuscript).

REVIEWERS' COMMENTS:

Reviewer #1 (Remarks to the Author):

The authors have addressed my comments well and the manuscript has improved.

Reviewer #2 (Remarks to the Author):

The authors have addressed the concerns raised by the reviewers.